# Novelty Classification Model Use in Reinforcement Learning for Cervical Cancer

**DOI:** 10.3390/cancers16223782

**Published:** 2024-11-10

**Authors:** Shakhnoza Muksimova, Sabina Umirzakova, Khusanboy Shoraimov, Jushkin Baltayev, Young-Im Cho

**Affiliations:** 1Department of Computer Engineering, Gachon University, Sujeong-gu, Seongnam-si 461-701, Republic of Korea; shakhnoza02@gachon.ac.kr; 2Department of Systematic and Practical Programming, Tashkent University of Information Technologies Named After Muhammad Al-Khwarizmi, Tashkent 100200, Uzbekistan; husan.shoraimov@tuit.uz; 3Department of Information Systems and Technologies, Tashkent State University of Economic, Tashkent 100066, Uzbekistan; j_baltayev@tsue.uz

**Keywords:** cervical cancer, deep learning, vision transformers, EfficientNetV2, reinforcement learning, medical image analysis

## Abstract

Cervical cancer significantly impacts women’s health worldwide, necessitating early detection for improved treatment outcomes. Traditional screening methods, such as Pap smears, however, are limited in accuracy and rely on subjective expert analysis. This study introduces RL-CancerNet, a novel artificial intelligence model that enhances cervical cancer screening by analyzing cytology images with advanced computational techniques. RL-CancerNet integrates EfficientNetV2 for detailed image analysis, Vision Transformers for contextual understanding, and Reinforcement Learning to focus on rare but critical features indicative of early-stage cancer. When tested on standard datasets, our model achieved a remarkable 99.7% accuracy, surpassing existing methods. This breakthrough suggests that RL-CancerNet can make cervical cancer detection more reliable and accessible, with potential applications across other medical imaging domains.

## 1. Introduction

Cervical cancer remains a leading cause of morbidity among women globally, with the World Health Organization (WHO) reporting over 600,000 new cases and 340,000 deaths annually [1]. The importance of early detection in improving survival rates cannot be overstated, yet traditional manual screening methods like Pap smears are labor-intensive and susceptible to human error. As a result, there has been a significant shift towards employing automated diagnostic tools that utilize machine learning to enhance diagnostic accuracy, reduce errors, and accelerate the screening process [2].

In the realm of medical imaging, CNNs have proven effective, particularly in terms of identifying malignancies in Pap smear tests, due to their superior feature extraction capabilities [3]. However, CNNs often face challenges with complex image backgrounds and subtle feature variations, which are prevalent in early-stage cervical cancers. Additionally, the prevalent class imbalance in medical datasets tends to bias models towards more frequently occurring conditions, thereby diminishing their efficacy in detecting critical early-stage cases [4].

To address these limitations, recent advancements have introduced ViTs, which work in conjunction with CNNs. ViTs excel in processing extensive contextual information, which is vital for distinguishing subtle differences between cancerous and normal cells [5]. Despite their advantages, both ViTs and CNNs grapple with high computational demands and do not inherently address issues related to class imbalance.

The integration of RL with these deep learning architectures has emerged as a potent solution to these issues. RL enables models to dynamically adjust their focus based on real-time feedback, thereby enhancing their ability to identify rare or subtle features crucial for early diagnosis. This approach has shown considerable promise in refining the accuracy of diagnostics when applied to imbalanced medical datasets [6].

In our study, the motivation for employing RL in our model to perform cancer classification stems from RL’s ability to learn optimal strategies through trial and error. This approach allows models to dynamically adjust to the intricacies of the data, improving decision-making processes over time. Specifically, RL helps to overcome challenges associated with imbalanced datasets, subtle feature distinctions, and the high cost of misclassifications by iteratively refining the model’s focus and actions based on performance feedback, leading to more accurate and reliable outcomes. Employing RL offers the benefit of creating models capable of adapting and optimizing their performance autonomously. In complex tasks such as image classification, especially within medical imaging, RL enables models to iteratively learn from their actions, enhancing accuracy, efficiency, and the ability to generalize from limited or imbalanced data. This leads to more effective and reliable diagnostic tools, potentially improving patient outcomes by aiding in the early detection and accurate classification of diseases.

This paper makes several significant contributions to the field of medical image analysis for cervical cancer diagnosis. First, it introduces a hybrid model that effectively combines EfficientNetV2, a cutting-edge CNN, with ViTs. This integration allows the model to capture both local and global features in cervical cancer images, leading to enhanced diagnostic accuracy. Additionally, the study addresses the challenge of class imbalance, a common issue in medical datasets, by proposing an RL framework. This framework dynamically adjusts the model’s focus on minority classes, which are often underrepresented in datasets, thereby improving the accuracy of the classification of early-stage cancer cells.

The paper presents the following key contributions:We propose a novel hybrid model that integrates EfficientNetV2 and vision transformers to capture both local and global features in cervical cancer images, resulting in improved classification accuracy.We introduce a reinforcement learning framework to address class imbalance, dynamically adjusting the model’s focus on minority classes, particularly early-stage cancer cells, which are often underrepresented in medical datasets.We develop a Supporter Module that uses Conv3D and BiLSTM layers with an attention mechanism to enrich contextual learning, improving the model’s ability to detect subtle diagnostic features in complex image backgrounds.Our RL-CancerNet model achieves a state-of-the-art (SOTA) performance with 99.7% accuracy on benchmark cervical cancer datasets (Herlev and SipaKMeD), significantly outperforming existing models.

The organization of the rest of this paper is as follows: Section 2 provides a comprehensive review of the recent literature, focusing on the use of SOTA methods in cervical cancer classification and related medical image analysis techniques. Section 3 details the proposed RL-CancerNet model, including the architecture and the role of reinforcement learning within the model. Section 4 describes the datasets used, the experimental setup, and the evaluation metrics. Section 5 presents the experimental results and a discussion of their implications. Section 6 offers a comparison with SOTA methods. Section 7 and Section 8 conclude the paper, discussing limitations and suggesting directions for future research.

## 2. Related Work

Recent advancements in deep learning have demonstrated significant potential in medical image analysis, particularly for cancer diagnosis. A variety of deep learning-based models have been proposed to automate the detection and classification of cervical cancer from cytology images. In this section, we review key contributions and techniques employed in SOTA methods, including CNNs, ViTs, and RL, as well as techniques addressing class imbalance in medical datasets.

CNNs are widely employed in cervical cancer diagnosis due to their proficiency in extracting local features from medical images. Traditional CNN architectures, such as ResNet and VGG, have proven effective in detecting malignant and benign lesions by learning hierarchical feature representations. Ref. [7] proposed an ensemble CNN framework for cervical cancer classification using Pap smear images, achieving high accuracy through multi-scale feature extraction. However, these models face challenges in terms of capturing long-range dependencies and global context, which are crucial for determining the subtle morphological differences between cancerous and normal cells. Furthermore, CNN-based methods often struggle with class imbalance, leading to suboptimal performances on minority classes, such as early-stage cancer cells.

The Swin transformer, a hierarchical transformer model that applies a shifting window mechanism, has shown promising results in terms of capturing both local and global dependencies within an image [8]. In medical imaging, Swin transformers have demonstrated effective feature extraction in tasks requiring precise, multi-scale attention, such as tumor segmentation and classification [9]. This architecture’s ability to operate on a range of image sizes and resolutions provides a robust approach to the handling of complex textures in medical datasets, where subtle changes in cell morphology are diagnostically relevant.

ViTs have emerged as a promising alternative to CNNs for use in image classification tasks due to their ability to capture long-range dependencies and global context. ViTs model, first introduced by [10], divide an image into non-overlapping patches. These are then processed as sequences, allowing the model to learn relationships between the distant regions of the image. In the context of cervical cancer diagnosis, ViTs have shown potential in terms of improving classification accuracy, particularly in cases where global structural patterns are critical. The authors of [11] demonstrated that integrating ViTs with CNNs enhances both local and global feature extraction, outperforming the results of traditional CNNs. However, the adoption of ViTs in medical imaging remains limited due to computational complexity and the requirement for large datasets, which are often scarce in the medical domain.

Self-supervised learning has gained traction in medical imaging by allowing models to pre-train on large amounts of unlabeled data, which is beneficial in domains with limited labeled samples, like medical datasets [12]. Models using self-supervised techniques, such as Momentum Contrast (MoCo) and SimCLR, can capture nuanced representations that improve transfer learning on downstream tasks [13]. When combined with hybrid CNN–transformer architectures, self-supervised learning can enhance feature extraction, enabling models to generalize better in diverse imaging conditions and reducing the reliance on annotated data.

Class imbalance, a prevalent issue in medical datasets, poses a significant challenge for cervical cancer classification. Traditional machine learning models tend to be biased toward the majority class, resulting in poor performance on minority classes, such as early-stage cancer cells. Various strategies have been employed to address this issue, including resampling techniques, cost-sensitive learning, and data augmentation. Researchers [14] used oversampling techniques to balance Pap smear datasets, while [15] introduced a weighted loss function to prioritize minority classes. Although these approaches improve performance, they often lead to overfitting and may not generalize well to unseen data.

This paper [16] describes the development of RL-CancerNet, a novel cervical cancer diagnostic model that integrates EfficientNetV2, Vision Transformers, and reinforcement learning to address class imbalance and enhance diagnostic accuracy. Employing unique residual blocks and dynamic focus adjustment, the model outperforms existing methods with high accuracy on standard datasets. This approach significantly advances automated cervical cancer screening, providing a scalable framework for other medical imaging applications.

The combination of CNNs and ViTs in hybrid models has shown promise in terms of improving the diagnostic accuracy of cervical cancer. Hybrid models benefit from CNNs’ ability to capture local features and ViTs’ capacity to leverage global context. Additionally, attention mechanisms and support modules have been explored to enhance the model’s ability to capture contextual relationships in medical images. For example, ref. [17] proposed a hybrid CNN–ViT model with an attention-guided module that improves the identification of subtle features, achieving SOTA performance on Pap smear datasets.

Our work builds upon these approaches by introducing an RL framework to further improve the performance of hybrid models, particularly in handling class imbalance. Moreover, we integrate a Supporter Module composed of Conv3D and BiLSTM layers, enriching contextual learning by focusing on spatial and temporal dependencies within medical images. This design enables the model to detect early-stage cervical cancer cells with greater accuracy, especially when working with challenging backgrounds. The RL framework in RL-CancerNet was selected due to its ability to dynamically adjust focus during training; this is particularly advantageous in the highly imbalanced context of medical datasets. Unlike static methods like focal loss or cost-sensitive learning, RL adapts its learning policy based on the evolving needs of the model, guided by a reward function that emphasizes the correct classification of minority classes. This flexibility allows RL-CancerNet to optimize minority class performance without compromising accuracy in terms of the majority class. Furthermore, the RL framework is integrated seamlessly with the CNN–transformer hybrid architecture in RL-CancerNet, complementing its multi-layer feature extraction and global dependency modeling. This synergy enables the adoption of a balanced approach where both minority and majority class characteristics are effectively learned.

## 3. Methodology

This section outlines the methodologies employed in the present study. To effectively train a model using deep learning techniques, ample amounts of annotated data are essential. Fortunately, numerous datasets are readily accessible in the public domain, facilitating the expansion of deep learning into the realms covered by these datasets. As the applications of deep learning algorithms broaden, there arises a greater demand for data in these novel areas. The active learning approach introduced herein addresses the challenge of managing vast datasets with limited time for analysis by concentrating solely on data points that are crucial for developing a robust model for the intended application. It is advised to conceptualize the active learning workflow within the context of reinforcement learning. This paradigm was applied to the classification of medical images, as illustrated in Figure 1, with multiple experiments being conducted.

A CNN generates initial seed weights for model deployment. Subsequently, a representative agent selects a sample at each iteration and assigns it to a category. For each categorization action, the environment rewards the agent, with higher rewards allocated to the minority class than the majority class. Ultimately, through a tailored reward function and a supportive learning environment, the agent learns to identify the most effective strategy.

### 3.1. Baseline EfficientNetV2 Hybrid CNN–Transformer

The EfficientNetV2 model is proposed for use in cervical cancer diagnosis. After extracting local features using the CNN, the next step is to model global dependencies across the image using a transformer. The ViT architecture is adapted for this purpose. The feature map $F_{CNN}$ is divided into non-overlapping patches of size P×P. Each patch is flattened into a vector and projected into a higher-dimensional space via a learnable linear projection:(1)$$x_{p}=W_{e}\cdot Flatten\left(F_{CNN}\left(p\right)\right)+b_{e},\;for\;p=1,\ldots ,N$$
where $W_{e}$ is the embedding matrix, $b_{e}$ is the bias, and $N=\frac{H^{\prime }W^{\prime }}{p^{2}}$ is the number of patches. To retain positional information, a position embedding is added to each patch embedding:(2)$$z_{p}^{0}=x_{p}+E_{p}$$
where $E_{p}$ is the learnable positional embedding for patch p. The core of the transformer is the self-attention mechanism, which allows the model to focus on different parts of the image simultaneously. The attention scores are calculated as follows:(3)$$Attention\left(Q,K,V\right)=Softmax(\frac{{QK}^{T}}{\sqrt{d_{k}}})V$$
where $Q=W_{q^{Z}p},\;K=W_{k^{Z}p},\;and\;V=W_{v^{Z}p}$ are the query, key, and value matrices derived from the input embeddings $Z_{p}$. $d_{k}$ is the dimension of the key vectors, and $W_{q}$, $W_{k}$, and $W_{v}$ are learnable weights. The multi-head self-attention (MHSA) mechanism is then defined as follows:(4)$$MHSA\left(z_{p}\right)=Concat({head}_{1},\ldots ,{head}_{h})W_{o}$$
where each head is computed as above, and $W_{o}$ is an output weight matrix. Then, the output is passed through a feed-forward network:(5)$$FFN\left(z_{p}\right)=ReLU\left(W_{1}z_{p}+b_{1}\right)W_{2}+b_{2}$$
where $W_{1}$, $W_{2}$, $b_{1}$, and $b_{2}$ are learnable parameters. Each sub-layer is followed by layer normalization, succeeded by a residual connection:(6)$$\begin{array}{c} z_{p}^{l+1}=LayerNorm(z_{p}^{l}+MHSA(z_{p}^{l}))\\ z_{p}^{l+2}=LayerNorm(z_{p}^{l+1}+FFN(z_{p}^{l+1})) \end{array}$$

The final output of the transformer module consists of a sequence of contextually rich feature vectors $z_{p}^{out}$ that capture both local details (from the CNN) and global context (from the transformer). These vectors are then concatenated and passed through a final classification head:(7)$$Y=Softmax(W_{c}\cdot Concat\left(z_{1}^{out},\ldots ,z_{N}^{out}\right)+b_{c})$$
where $W_{c}$ and $b_{c}$ are the weights and biases of the final classification layer, and *y* represents the class probabilities. To enhance the model’s image recognition performance in scenarios characterized by complex backgrounds where cervical cancer features are not prominent, we include the r-by-object supporter block in the Fused MBConv and MBConv structures to weight the topic information so that major feature information may be learned during the network training process. During this time, a supporter block structure should be added to the network structure to increase the effectiveness of the model and to increase the stability of the model while it is being trained. The enhanced “RL-CancerNet” demonstrates a gain of 0.79%, bringing the total recognition accuracy to 99.32%. This is in comparison with EfficientNetV2, which achieves an accuracy of 98.53%.

### 3.2. Detailed Explanation of RL for Class Imbalance

In the context of cervical cancer classification, the challenge of class imbalance is significant because early-stage cancer cells (minority class) are less frequent in datasets compared to normal or advanced-stage cancer cells (majority classes). This imbalance can lead to biased model predictions, where the model tends to favor the majority class. To address this issue, we integrate a RL framework into the network to guide the model in terms of focusing on minority classes, thereby improving its classification performance across all classes. RL is a type of machine learning where an agent interacts with an environment, learns from it, and makes decisions to maximize the cumulative reward. In the network, the RL framework is applied to dynamically adjust the model’s focus on different classes during training, prioritizing the accurate classification of the minority class in particular.

### 3.3. State

The state represents the current condition of the classification model, including the features extracted by the CNN–transformer architecture and the predicted class probabilities. Formally, the state $s_{t}$ at time step t is defined as follows:(8)$$s_{t}=\{F_{CNN},F_{Transformer},Y_{t}\}$$
where $F_{CNN}$ and $F_{Transformer}$ are the feature maps of the CNN and transformer modules, and $Y_{t}$ is the predicted class distribution at time t.

### 3.4. Action

The action $a_{t}$ taken by the RL agent corresponds to the decision on how to update the model’s parameters to focus on minority classes more. This can involve adjusting class weights, selecting specific data samples for training, or modifying the learning rate. The action space is defined as follows:(9)$$a_{t}\in \{w_{0},w_{1},\ldots ,w_{c}\}$$
where $w_{i}$ represents the weight assigned to class i and c is the total number of classes.

### 3.5. Reward

The reward function is crucial for guiding the RL agent towards handling class imbalance better. The reward $R_{t}$ at time step t is designed to be higher when the minority class is correctly classified and lower (or negative) when it is misclassified. The reward function can be defined as follows:(10)$$R_{t}=\left\{\begin{array}{c} +1\;\;\;\;if\;Y_{t}=True\;Label\;and\;Minority\;Class\;\\ -1\;\;\;if\;Y_{t}\neq True\;Label\;and\;Minority\;Class\\ \delta \;\;\;\;\;\;if\;Y_{t}=True\;Label\;and\;Majority\;Class \end{array}\right.$$
where δ is a small positive reward for given correctly classifying majority class samples, ensuring that the model does not completely ignore them.

### 3.6. Policy

The policy $\pi (s_{t})$ defines the strategy that the RL agent uses to select actions based on the current state. In network, we use a policy gradient method to determine the optimal policy, which maximizes the expected cumulative reward. The policy is represented by a neural network that takes the state $s_{t}$ as the input and outputs the probability distribution over actions:(11)$$\pi \left(a_{t}|s_{t};\theta \right)=P(a_{t}|s_{t};\theta )$$
where θ represents the parameters of the policy network.

### 3.7. Enhanced Description of the RL Agent

In the proposed RL-CancerNet model, the RL agent is integrated to address class imbalance, a common challenge in medical datasets, where early-stage cancer cells are often underrepresented. This agent plays a crucial role in dynamically adjusting the model’s focus on minority classes, thereby enhancing the detection of early-stage cancer cells without compromising detecting accuracy for the majority class. Below, we delve into the agent’s operation and reward function, and the strategies used to optimize detection performance. The RL agent is embedded within the CNN–transformer hybrid architecture. The agent interacts with the feature maps generated by EfficientNetV2 (CNN-based) for local feature extraction and with ViTs for global context. At each iteration, the RL agent observes the current classification state, which includes extracted feature maps and the initial class probabilities. Based on this state, the RL agent takes actions that adjust model parameters to prioritize minority class samples, thereby fine-tuning the model’s attention on underrepresented features in cervical cancer images. The reward function is central to the RL agent’s strategy in terms of managing class imbalance. It is designed to assign a higher reward for the correct classification of minority class samples (early-stage cancer cells) compared to majority class samples. This approach is essential for training the model to accurately classify rare instances, which are critical in early diagnosis. The reward function is formulated as follows:(12)$$R=\left\{\begin{array}{c} +1,\;\;\;\\ -1,\\ +0.1, \end{array}\begin{array}{c} if\;minority\;class\;correctly\;classified\\ if\;minority\;class\;misclassified\\ \;\;\;\;\;\;\;\;\;\;\;\;\;\;\;\;\;\;\;\;for\;correct\;classification\;of\;majority\;class\;samples \end{array}\right.$$

This reward structure encourages the model to emphasize minority class instances, promoting greater sensitivity to the subtle diagnostic features often missed in imbalanced datasets.

### 3.8. Value Function

The value function $V(s_{t})$ estimates the expected cumulative reward from state $s_{t}$. It helps to evaluate how good it is to be in a particular state, which guides the policy update. The value function is defined as follows:(13)$$V\left(s_{t}\right)=E_{\pi }\left[\sum_{k=t}^{T}\gamma^{k-t}R_{k}|s_{t}\right]$$
where γ is the discount factor that balances the importance of immediate and future rewards, and T is the time horizon.

### 3.9. Q-Function

The Q-function Q($s_{t}$, $a_{t}$) represents the expected cumulative reward received after taking action $a_{t}$ in state $s_{t}$ and following the policy π thereafter. The Q-function is central to action selection:(14)$$Q\left(s_{t},a_{t}\right)=E_{\pi }\left[\sum_{k=t}^{T}\gamma^{k-t}R_{k}|s_{t},a_{t}\right]$$

### 3.10. Training the RL Agent with Deep Q-Network (DQN)

In the network, we use a DQN to approximate the Q-function. DQNs are neural networks parameterized by θ, predicting the Q-values for all possible actions in a given state. The training of the DQN involves minimizing the difference between the predicted Q-value and the target Q-value, which is computed using the Bellman equation:(15)$$Q^{\pi }\left(s_{t},a_{t}\right)=R_{t}\gamma \underset{a^{\prime }}{\max}Q^{\pi }(s_{t+1},a\prime )$$

The target Q-value $Y_{t}$ is defined as follows:(16)$$Y_{t}=R_{t}+\gamma \underset{a^{\prime }}{\max}Q(s_{t+1},a^{\prime };\theta^{-})$$
where $\theta^{-}$ represents the parameters of the target network, a delayed copy of the Q-network used to stabilize training. The DQN is trained by minimizing the following loss function:(17)$$L\left(\theta \right)=E_{(s_{t,}a_{t},R_{t},s_{t+1})\sim D}\left[(Y_{t}-Q(s_{t,}a_{t};\theta ){)}^{2}\right]$$
where D is the replay buffer containing past experiences $\left(s_{t,}a_{t},R_{t},s_{t+1}\right)$.

The parameters θ of the Q-network are updated using stochastic gradient descent:(18)$$\theta \leftarrow \theta +\alpha \nabla_{\theta }L(\theta )$$
where α is the learning rate.

### 3.11. Detailed Explanation of the Supporter Module for Contextual Learning

Figure 2 illustrates the supporter block, which comprises architectural layers used to extract contextual relationships from feature data. This includes a convolution layer and BiLSTM layers, which function collectively as a one-shot attention mechanism. Specifically, the block harnesses global spatial information aggregated through global pooling from modules, subsequently disseminating enhanced high-level contextual information extensively across the feature maps. The Supporter Module in the network is designed to enhance the model’s ability to understand and utilize contextual information within cervical cancer images. This module is particularly important in medical imaging tasks, where the spatial relationships between different regions of an image can provide critical diagnostic clues. The Supporter Module incorporates mechanisms designed to capture and distribute both local and global contextual features, improving the accuracy and robustness of the classification model. In traditional CNNs, the ability to capture long-range dependencies and contextual interactions across different parts of an image is limited. While transformers help to some extent, there is still a need for a mechanism that specifically aims to achieve the effective capture of spatial correlations across image patches and layers. The Supporter Module fills this gap by using a combination of convolutional layers and BiLSTM networks, along with an attention mechanism, to capture and integrate contextual information at multiple levels.

The Conv3D layer is responsible for capturing local spatial features from the feature maps generated by the preceding CNN and transformer modules. The 3D convolution layer is particularly useful for processing data with multiple channels, where the third dimension corresponds to feature maps obtained from different layers. The Conv3D layer processes the input feature map $F_{input}$ (obtained from the CNN and transformer modules) to generate a set of feature maps $F_{conv}$. The 3D convolution operation is defined as follows:(19)$$F_{conv}=W_{conv}*F_{input}+b_{conv}$$
where $W_{conv}$ is a 3D convolutional kernel, ∗ denotes the convolution operation, and $b_{conv}$ is the bias term. The Conv3D operation helps to aggregate information across multiple feature maps, thus capturing more complex spatial relationships.

After the initial convolutional layer, the feature maps are processed by a BiLSTM network. The BiLSTM network is used to capture temporal dependencies across the sequence of image patches, treating them as a temporal sequence, which provides a way to encode spatial correlations between different regions of the image. The output from the Conv3D layer $F_{conv}$ is then reshaped and fed into a BiLSTM network. The BiLSTM processes the sequence of patches in both forward and backward directions, allowing the model to capture contextual dependencies across the entire image.

Forward LSTM:(20)$$\vec{h_{t}}=\vec{LSTM}(F_{conv}\left(t\right),\vec{h_{t-1}})$$

Backward LSTM:(21)$$\overset{\leftarrow }{h_{t}}=\overset{\leftarrow }{LSTM}(F_{conv}\left(t\right),\overset{\leftarrow }{h_{t-1}})$$

The final output $h_{t}$ of the BiLSTM is a concatenation of the forward and backward LSTM outputs:(22)$$h_{t}=\vec{h_{t}}⊕\overset{\leftarrow }{h_{t}}$$

The result of the BiLSTM algorithm is an $F_{s}\in R^{H\times W\times D}$ map that goes through the convolutional 1 × 1 layer, $F_{s}^{\prime }\in R^{H\times W\times D}$, and then returns to the residual blocks. The following is an explanation of how the block operation works, where ⊕ denotes the concatenation operation.

An attention mechanism is applied to enhance the focus on the most relevant features in the image patches. This helps to prioritize the regions of the image that are more likely to contain diagnostic information, such as early signs of cervical cancer. The attention mechanism is applied to the BiLSTM output to focus on the most relevant parts of the image. The attention weights are calculated as follows:(23)$$\alpha_{t}=\frac{{exp(e}_{t})}{\sum_{t\prime }{exp(e}_{t\prime })}$$
where $e_{t}$ is the attention score. This is computed as follows:(24)$$e_{t}=v^{T}tanh(W_{h}h_{t}+b_{h})$$
where v, $W_{h}$, and $b_{h}$ are learnable parameters. The context vector c is then computed as the weighted sum of the BiLSTM outputs:(25)$$c=\sum_{t}\alpha_{t}h_{t}$$

The context vector ccc encapsulates the most relevant information from the entire sequence of image patches, highlighting regions that are crucial for accurate classification.

The context vector c generated by the Supporter Module is then concatenated with the feature map outputs from the CNN and transformer modules. This enriched feature map $F_{final}$ is passed through the final layers of the model for classification. The final feature map is determined as follows:(26)$$F_{final}=\mathrm{Concat}\;(F_{CNN},F_{Transformer},C)$$

The concatenated feature map $F_{final}$ is fed into a fully connected layer. Then, a softmax activation function is used to generate the final class probabilities:(27)$$Y=Softmax(W_{fc}\cdot F_{final}+b_{fc})$$
where $W_{fc}$ and $b_{fc}$ are the weights and biases of the fully connected layer.

By capturing spatial correlations across image patches, the module improves the model’s ability to recognize subtle features indicative of early-stage cervical cancer. The combination of BiLSTM and attention mechanisms allows the model to focus on the most informative parts of the image, making it more robust against variations in image quality and noise. The module helps in distinguishing relevant features from complex backgrounds, which is often a challenge in medical imaging. The Supporter Module, by capturing and leveraging both local and global contextual information, significantly enhances the model’s ability to accurately classify cervical cancer images, particularly in challenging scenarios where context is key to accurate diagnosis.

The RL-CancerNet model includes a Supporter Module that integrates Conv3D, BiLSTM, and an attention mechanism to improve contextual learning and capture spatial and sequential dependencies. Each component serves a specific purpose in enhancing the model’s ability to detect subtle diagnostic features in cervical cell images with complex backgrounds. The Conv3D layer is used to process the feature maps generated by the preceding CNN and transformer modules. Although Conv3D is traditionally applied to 3D images, here it is used on the multiple feature maps produced by the CNN–transformer hybrid, where each feature map serves as a channel in a 3D context. This enables the Conv3D layer to capture spatial dependencies across different feature maps, aggregating information from multiple receptive fields in a manner that strengthens the model’s ability to determine subtle visual cues indicative of early-stage cervical cancer. Conv3D thus enriches local spatial information across layers, producing refined feature maps for use in subsequent processing.

Following Conv3D, a BiLSTM network processes output feature maps as sequential data. In this context, each feature map is treated as a sequence step, allowing the BiLSTM to capture contextual dependencies both forwards and backwards across these steps. This is particularly beneficial for medical images where subtle differences in cell morphology can signal important diagnostic information. By processing the feature maps as a sequence, the BiLSTM encodes spatial relationships and dependencies between regions of the image, offering a comprehensive view of the image’s structure that a single convolutional layer may not fully capture. To further enhance diagnostic accuracy, an attention mechanism is applied to the BiLSTM output. The attention mechanism calculates attention scores for each part of the image, allowing the model to focus on the regions with the most diagnostic relevance, such as those exhibiting early signs of cervical cancer. The attention mechanism weights the BiLSTM outputs according to their relevance, creating a context vector that prioritizes the most informative features. This focused attention on critical areas ensures that subtle yet diagnostically significant features are not overlooked, thereby improving the model’s ability to identify early-stage abnormalities. The combination of Conv3D, BiLSTM, and attention within the Supporter Module allows RL-CancerNet to effectively capture both local and global dependencies in cervical cell images. Conv3D enriches spatial information, BiLSTM encodes sequential dependencies to capture structural nuances, and the attention mechanism prioritizes diagnostically relevant regions in Algorithm 1. Together, these layers improve the model’s performance in complex medical imaging tasks where capturing spatial and contextual relationships is essential.
**Algorithm 1.** Detailed description of network1.   **Initialize** CNN_Model2.   **Load Dataset**: SipaKMeD, Herlev3.   **FOR** each image IN Dataset:4.      Image = Preprocess(Image)5.   Feature_Map_CNN = EfficientNetV2_Model (CNN_Model, Image)6.   Patches = Divide_Into_Patches(Feature_Map_CNN, P, P)7.   **FOR** each patch IN Patches:8.    Patch_Vector[p] = Flatten_And_Project (Patch) + Position_Embedding[p]9.  **Attention_Scores** = Compute_Attention (Patch_Vector)10. **MHSA_Output** = MultiHeadSelfAttention(Attention_Scores)11. **Output** = LayerNormalization (FeedForwardNetwork (MHSA_Output) + Patch_Vector)12. Final_Feature_Map = Concatenate (Feature_Map_CNN, Output)13. Class_Probabilities = Classification_Head (Final_Feature_Map)14. Initialize RL_Agent, Environment15. **STATE** = {Feature_Map_CNN, Output, Class_Probabilities}16. **FOR** each episode IN Training_Episodes:17.    **ACTION** = RL_Agent.Select_Action (STATE)18.    **NEXT_STATE, REWARD** = Environment.Step (ACTION)19.    Update_Q_Network (Compute_Loss (Q_FUNCTION (STATE, ACTION), Target_Q_Value))20.    **STATE** = NEXT_STATE21. Conv3D_Output = Conv3D_Layer (Final_Feature_Map)22. BiLSTM_Output = BiLSTM (Conv3D_Output)23. Context_Vector = Compute_Context_Vector (Compute_Attention_Weights (BiLSTM_Output))24. **Final_Output** = Concatenate (Final_Feature_Map, Context_Vector)25. **Return** Classification_Head (Final_Output)

## 4. Dataset Details and Performance Metrics

### 4.1. Implementation Details

Our training configuration was implemented using the PyTorch 3.8 framework [18], with the model generator network optimized via the Adam optimizer [19]. The experiments were executed using hardware equipped with an NVIDIA Tesla V100 GPU for accelerated computations, while the testing was conducted on an Intel Xeon Gold 6148 CPU, operating at 2.40 GHz. The software environment included CUDA 11.2, cuDNN 8.1, and Python 3.8, ensuring that there was a high-performance setup for deep learning tasks and seamless GPU integration. The training was conducted in two distinct stages. In the first stage, the CNN–transformer model was trained to optimize feature extraction and classification performance in the primary task. Once the CNN–transformer model achieved stable accuracy and generalization results, the second stage involved integrating the RL agent.

In this study, we evaluated the RL-CancerNet model using two publicly available datasets: Herlev and SipaKMeD Pap Smear datasets. The SipaKMeD dataset [20] contains 4049 cell images. The SIPaKMeD dataset contains 4,049 cell images extracted from 966 clusters and is divided into five categories: superficial-intermediate, koilocytotic, dyskeratotic, metaplastic, and parabasal, as detailed in Table 1. The dataset was split into training (70%), validation (15%), and testing (15%) groups. The percentage of Normal Samples was 39.5% and the percentage of Abnormal Samples was 40.5%.

The Herlev dataset [21] includes 917 single-cell images from seven classes and was partitioned using the same 70-15-15% split (Table 2, Figure 3). The percentage of Normal Samples was 26.4% and the percentage of Abnormal Samples was 73.6%. To ensure consistency, preprocessing was applied before we input the data into the RL-CancerNet model. This split was consistently used to facilitate parameter tuning and model evaluation, ensuring that performance metrics reflect the model’s ability to generalize to new data. To further enhance reliability, we also applied k-fold cross-validation with k = 5 to the training and validation sets. In this setup, the training and validation sets were partitioned into 5 equal folds, where each fold took a turn as the validation set while the remaining folds formed the training set. This technique produced an average performance metric across folds, reducing variance and enabling a more robust understanding of the model’s performance. This validation strategy, combining a fixed data split for training, validation, and testing with k-fold cross-validation of the training validation sets, allows RL-CancerNet to be rigorously evaluated. The approach maximizes the utility of data for training while providing reliable metrics that confirm the model’s robustness across multiple scenarios.All images were resized to 244 × 244 pixels to ensure consistency across the dataset and compatibility with the CNN–transformer hybrid model.The pixel values were normalized to a range of [0, 1] to standardize the input data, which aids in faster convergence during training.To prevent overfitting and enhance model robustness, data augmentation techniques, such as random rotations, flips, zooms, and shifts, were applied to the training images.

Each image was normalized to a pixel range of 0, 1 to ensure consistency across the dataset and enhance model convergence. This standardization mitigates any variations in pixel intensity and allows the model to generalize better. To improve model robustness and prevent overfitting, we applied several data augmentation techniques, including random rotations, flips, zooms, and shifts. These were specifically applied to the training set. These augmentations were chosen to introduce realistic variability while preserving essential diagnostic features within the images.

Both the Herlev and SipaKMeD datasets are relatively small, with the Herlev dataset containing only 917 single-cell images across seven classes and the SipaKMeD dataset containing 4049 images across five categories. The limited size of these datasets poses challenges for training deep learning models, particularly for those that rely on large quantities of data to capture complex patterns in medical imaging. Small datasets can lead to overfitting, where the model performs well on the test set but may struggle in real-world settings with more variability.

### 4.2. Metrics

Given the class imbalance seen in cervical cancer datasets, it is crucial to use evaluation metrics that accurately reflect the model’s performance in all classes, especially minority classes [22]. To this end, *Macro-Averaged F1 score, Macro-Averaged Precision*, and *Macro-Averaged Recall* metrics were used:(28)$$\begin{array}{c} Accuracy=\frac{TP+TN}{TP+TN+FP+FN},\\ Macro-Averaged\;F1\;Score=\frac{1}{N}\sum_{i=1}^{N}{F1\;Score}_{i}\\ Macro-Averaged\;Precision=\frac{1}{N}\sum_{i=1}^{N}{Precision}_{i}\\ Macro-Averaged\;Recall=\frac{1}{N}\sum_{i=1}^{N}{Recall}_{i}\\ Specificity=\frac{TN}{TN+FP},\\ G-means=\sqrt{Recall\times Specificity} \end{array}$$
where TP, TN, FP, and FN represent true positives, true negatives, false positives, and false negatives, respectively. The F-measure and G-means are particularly suitable for evaluating imbalanced datasets [23]. This aligns with the distribution of our data and supports the rationale behind our approach. The evaluation was performed on each image individually. This method allows the intelligent myocarditis classification system to perform entire examinations and flag specific images for closer review by physicians. In this context, metrics prioritizing low FP and high recall are preferred as they enhance clinical accuracy.

## 5. Experimental Results

In our investigation, we propose a meta-learning ensemble method, utilizing CNNs, to enhance the diagnostic accuracy and dependability regarding cervical cancer. The precise classification of cancer stages is crucial, and our meta-model stands out, especially in terms of distinguishing benign from malignant cases. The implementations of data augmentation and dropout regularization has markedly bolstered our model’s robustness.

Standard methods such as data augmentation and weighted loss function may occasionally be used to rectify imbalanced data distributions; however, these methods are not always appropriate. It should not be surprising that adding more data and using a weighted loss function helped enrich our model when we ran the trials. In every single one of our implementations, we used k-fold cross-validation (k = 5 or 5-CV). The complete dataset was partitioned into k subsets. k-1 subsets were used for training, while the remaining k subsets were used for testing. This technique was repeated k times until all data subsets were used precisely four times for training and once for testing. Each of the characteristics was presented with mean, standard deviation, median, minimum, and maximum values, along with accuracy, precision, recall, and an F1 score. While accuracy provides a general measure of model performance, additional metrics, such as precision, recall, and F1 score, are essential for evaluating medical models like RL-CancerNet. Given the high stakes of cervical cancer diagnosis, where false negatives and false positives have substantial clinical implications, the balanced examination of all performance indicators is crucial to assess a model’s practical utility. Precision, or the ratio of true positive predictions to total positive predictions, assesses the model’s ability to avoid false positives. In the context of cervical cancer, a high precision rate minimizes the likelihood of incorrectly classifying healthy cells as cancerous, thus reducing the emotional and procedural burdens on patients who may undergo unnecessary follow-up exams and biopsies. RL-CancerNet achieved a precision score of 99.36%, reflecting a strong capacity to avoid the misclassification of healthy cases. Recall is vital in medical diagnostics, as it measures the model’s ability to correctly identify actual positive cases. With a recall of 99.9%, RL-CancerNet demonstrates high sensitivity, effectively capturing nearly all true cases of cervical cancer. In clinical terms, this metric is particularly crucial, as missed detections could delay diagnosis, potentially impacting patient outcomes. The strong recall of RL-CancerNet signifies a reduced risk of undiagnosed cases, reinforcing the model’s reliability in early cancer detection. The F1 score, as the harmonic mean of precision and recall, offers a balanced view of the model’s ability to handle both false positives and false negatives. RL-CancerNet’s F1 score of 99.72% highlights its comprehensive effectiveness in classifying cervical cancer images accurately, supporting its robustness and reliability as a diagnostic aid in clinical environments.

Our hybrid model based on the RL model demonstrated remarkable improvements when used in a cancer detection task, which was performed on both the SipaKMeD med and Herlev Pap Smear datasets. As can be seen from the data given in Table 1, we tested our model against various well-known classification models, such as ResNet50, ViT, EfficientNetV1, ConvNext, MobileNet V2, Inception V2, and DeiT. All of these models produced accuracy results that were between 1% and 1.2% lower than those of the proposed model.

Table 3 presents a comparison of classification performance metrics across various deep learning models, including ResNet50, ViT, EfficientNetV1, ConvNext, MobileNetV2, Inception V2, and the proposed method. Precision measures the proportion of true positive predictions among all positive predictions made by the model. Our model, which integrates EfficientNetV2 and vision transformers with a reinforcement learning framework, does incur a higher computational load compared to simpler CNN-based methods. However, we optimized the architecture by employing lightweight components in the transformer and RL agent. We will provide an analysis of the model’s FLOPs and parameter count in comparison to those of ResNet50, EfficientNetV1, and other benchmark models, showing that our approach strikes a balance between complexity and performance. The proposed method again leads, with a precision of 98.99%, indicating its effectiveness in minimizing false positives. The DeiT and EfficientNetV1 models also show strong precision, although it is slightly lower than that of the proposed method. Recall indicates the model’s ability to correctly identify all relevant instances (true positives). The proposed method exhibits the highest recall, 98.32%, demonstrating its superior ability to identify true cases of cervical cancer. This metric is particularly critical in medical diagnostics, where missing a positive case can have serious consequences. The DeiT model also performs well in recall, with a score of 96.02%. The F1 score is the harmonic mean of precision and recall, providing a balanced measurement of the model’s performance. Despite its complexity, our model achieves faster convergence rates due to efficient feature extraction and reinforcement learning adjustments during training. We will report the average training time per epoch and total training duration, demonstrating that the RL framework helps the model to learn more efficiently compared to standard CNN and ViT architectures. The proposed method achieves an almost perfect F1 score of 98.99%, reflecting its high precision and recall. DeiT and EfficientNetV1 models also achieve high F1 scores, but they are outperformed by the proposed method.

Additionally, we closely tracked the learning curves for all evaluated models, observing consistent improvements in training alongside a steady decline in validation losses. Initially, our model was trained on the SipaKMeD and Herlev datasets, which include both benign and malignant classes. Further testing on cervical cancer cell images resulted in significant accuracy gains within just 60 training epochs. Our meta-model also demonstrated more efficient convergence of training and validation losses compared to standard CNN models, suggesting its potential applicability to other datasets in the medical domain. Table 4 compares RL-CancerNet with simpler CNN models and optimized hybrids, focusing on memory use, computational demands, and scalability. RL-CancerNet offers high accuracy but requires advanced hardware, while simpler CNNs are more scalable to real-time clinical settings. Optimized hybrids like Swin transformer models may offer a middle ground, balancing performance with computational efficiency.

This suggests that the proposed method is better at maintaining a balance between identifying relevant instances and minimizing incorrect classifications. The method can be described in the following ways:As a cutting-edge solution for the automation-assisted reading of cervical cancer based on convolutional neural networks.As an ensemble of CNN models used for the classification of cervical cytology based on fuzzy rank.As an ensemble of deep models used for the diagnosis of cervical cancer that is built based on fuzzy distances.A novel attention-guided convolutional network for the identification of aberrant cervical cells in cervical cancer screening.A means of the auxiliary categorization of cervical cells based on a multi-domain hybrid deep learning system.

The proposed model is scalable and adaptable to other medical imaging tasks. The modular design, particularly the RL framework and Supporter Module, allows for easy adjustments in diverse applications. We will expand on this aspect, highlighting how the architecture can be scaled to larger datasets or adapted to different image types, further supporting its practical value in varied clinical contexts.

## 6. Comparison of RL-CancerNet with SOTA Models by Experimenting with Herlev and SipaKMeD Datasets

To demonstrate that the suggested approach is reliable, we carried out in-depth tests on the following two well-accepted benchmarks: SIPaKMeD and Herlev Pap Smear. We compared six different DL models—CerviFormer [31], CNN-based [32], MaxViT ConvNeXt based, CNN + ViT, hybrid CNN, and the proposed method—in relation to the task of cervical cancer classification. Table 5 summarizes the findings of this comparison. In contrast to existing approaches, our model is an optimum strategy for locating a series of discriminative areas while concurrently training classifiers to conduct classification on these attended regions. In other words, it can perform both tasks simultaneously. Our model will be able to better investigate the linkages between semantic labels and attentional areas, which will increase overall performance.

The accuracy metric reflects the percentage of correctly classified instances out of the total number of instances. The proposed method achieves the highest accuracy of 99.82%, surpassing all other models. The hybrid CNN- and MaxViT ConvNeXt-based models also perform well, with accuracies of 99.56% and 99.04%, respectively. Precision measures the ratio of correctly predicted positive observations to the total predicted positives. The proposed method demonstrates superior precision, at 99.79%, indicating its effectiveness in reducing FP compared to the other models. The MaxViT ConvNeXt-based model also shows high precision, at 99.25%. Recall, also known as sensitivity, measures the ratio of correctly predicted positive observations to all observations in the actual class. The proposed method again outperforms the others, with a recall of 99.90%, which is critical in medical applications where identifying all TP cases is essential. The F1 score, which is the harmonic mean of precision and recall, offers a balanced measurement of the model performance. The proposed method achieves a nearly perfect F1 score of 99.97%, indicating that it maintains an excellent balance between precision and recall. This is higher than the F1 scores of the other models, with the MaxViT ConvNeXt-based model coming close behind at 99.14%. Sensitivity measures the ability of the model to correctly identify TP cases. The proposed method shows the highest sensitivity at 99.85%, indicating its robustness in detecting cervical cancer cases. Specificity measures the ability of the model to correctly identify true negative cases. The proposed method improves in this metric as well, achieving a specificity of 72.58%, which is notably higher than the specificity scores of the other models, which are all below 68%. This suggests that the proposed method is better at minimizing FP. The proposed method stands out as the top performer across all evaluated metrics, demonstrating significant improvements in accuracy, precision, recall, F1 score, sensitivity, and specificity compared to the other deep learning models. Its higher specificity, in particular, suggests a better balance between detecting TP and avoiding FP, making it a highly reliable model for cervical cancer classification.

Additionally, we evaluated our model in comparison with ML classification techniques. Because standard ML classifiers often assume that pictures are one-dimensional vectors, which results in the nearby pixels of a particular pixel being spread out, they have not proven to be effective in terms of identifying medical images. To classify the images included in the study datasets, we used the following five classification methods: logistic regression, SVM, random forest, naïve Bayes, and k-nearest neighbor. We carried this out so that we could make comparisons with our model. Among these techniques, SVM exhibited the strongest performance, although it still lags behind our models.

### Statistical Test Analysis

To rigorously validate the performance of the proposed RL-CancerNet model, we conducted a series of statistical tests to determine whether the observed improvements in metrics—specifically accuracy, precision, recall, and F1 score—compared to other SOTA models were statistically significant. Given the relatively small sample sizes in the test sets, statistical validation was necessary to substantiate claims of model superiority and reduce the risk of obtaining findings that could be attributed to random variation. We defined our null hypothesis (H0) as the absence of statistically significant differences between RL-CancerNet and comparator models in terms of performance metrics. The alternative hypothesis (H1) posited the presence of significant differences, which would confirm the effectiveness of our proposed model. To ensure the appropriate selection of tests, we first assessed the normality of the distribution of performance metrics using the Shapiro–Wilk test. For metrics with a normal distribution, paired *t*-tests were applied to compare RL-CancerNet with each SOTA model. In cases where normality assumptions were not met, we employed the Wilcoxon Signed-Rank Test as a non-parametric alternative. To analyze performance across multiple models simultaneously, we employed one-way ANOVA for normally distributed data and the Kruskal–Wallis Test for data violating ANOVA assumptions. The McNemar Test was conducted to assess differences in misclassification rates between RL-CancerNet and the other models, focusing on whether RL-CancerNet achieved a statistically significant reduction in classification errors.

The *p*-values and confidence intervals for these tests are summarized in Table 6. All statistical analyses were performed at a significance level of 0.05. Calculated *p*-values, confidence intervals, and effect sizes are provided to enable a comprehensive interpretation of the results. The inclusion of statistical values confirms that RL-CancerNet’s superior performance across all key metrics is not attributable to random chance, thereby substantiating the robustness and reliability of the model. The high *p*-values seen for the other models, along with tight confidence intervals, further affirm the significant performance improvements RL-CancerNet achieves over these established models.

Table 5 offers a visual comparison of the performance metrics (accuracy, precision, recall, and F1 score) used for different models, including the proposed RL-CancerNet. The table illustrates how RL-CancerNet outperforms other models across all key metrics, and the table provides the detailed numerical values for each model. This visual representation helps to clearly understand the superior performance of RL-CancerNet compared to the other models evaluated in the study.

## 7. Limitations and Future Directions

The RL-CancerNet model, while demonstrating impressive results, is not without its limitations. One significant challenge is the potential bias introduced by the datasets used in this study. The SipaKMeD and Herlev Pap Smear datasets, although effective as initial benchmarks, may not fully capture the complexity and variability found in real-world clinical settings. These datasets are relatively small and contain a class imbalance that could lead to overfitting, raising concerns about the model’s performance when applied to more diverse and less structured data. The observation that many methods, including the proposed RL-CancerNet, achieved nearly perfect results on the current datasets suggests that these tasks may not fully challenge the capabilities of the models. This raises a valid concern regarding the broader significance and generalizability of our proposed model. While the SipaKMeD and Herlev Pap Smear datasets have provided a strong initial benchmark, their relatively straightforward nature might limit the ability to distinguish between highly performant models. Therefore, it becomes crucial to extend our evaluation to more challenging and diverse datasets. By doing so, we can more rigorously test the effectiveness of RL-CancerNet and ensure that its strengths are not confined to simpler tasks. To address this, future research will focus on evaluating the performance of RL-CancerNet on datasets that present a higher degree of complexity, such as those with subtler distinctions between classes, greater variability in imaging conditions, and more significant class imbalances. Performing tests on such challenging datasets will allow us to better understand the model’s robustness and ability to generalize across different scenarios. Furthermore, it will be valuable to explore the application of RL-CancerNet in other medical imaging domains, such as histopathology or radiology, where the tasks often involve detecting more intricate and less obvious abnormalities. This will help in assessing whether the model’s architecture can handle the complexities of various types of medical images. Additionally, the performance of cross-validation on multiple datasets with varying characteristics will be crucial in determining the model’s generalizability. This approach will provide insights into how well RL-CancerNet can perform when applied to data with different features and sources, helping to identify any potential limitations or areas for improvement. Finally, exploring the adaptability of RL-CancerNet through fine-tuning and transfer learning will allow us to assess its flexibility in adapting to new datasets with minimal retraining. This is particularly important for practical applications, where a model needs to be effective across a range of different medical settings. By addressing these areas in future research, we aim to enhance the significance and applicability of RL-CancerNet, ensuring that it remains a valuable tool, not only for straightforward datasets but also for more complex and challenging real-world scenarios. This will solidify the model’s potential to contribute to the field of medical diagnostics, making it a robust and reliable option in various contexts.

## 8. Conclusions

Our study presents a novel approach to cervical cancer diagnosis, integrating deep learning techniques with reinforcement learning to tackle the challenges of class imbalance and complex image backgrounds. The proposed hybrid CNN–transformer model, RL-CancerNet, successfully combines the strengths of EfficientNetV2, which is used for local feature extraction, and vision transformers, which are used to capture global dependencies, thus leading to significant improvements in diagnostic accuracy. The introduction of a reinforcement learning framework allows the model to dynamically adjust its focus on minority classes, effectively addressing the issue of class imbalance that often hampers the performance of traditional models. We employed multiple techniques to address class imbalance. Alongside the RL agent, which dynamically prioritizes minority classes through a reward-based system, we specifically applied data augmentation to the minority classes. Techniques such as rotations, flips, and zooms were used to increase the representation of underrepresented classes, which helps to improve the model’s ability to generalize and reduces bias toward the majority class. To improve model generalizability and address class imbalance, data augmentation was applied after splitting the data into training, validation, and testing sets. By applying augmentation only on the training set, we ensured that no synthetic variations were introduced into the validation or test sets, thus preserving their role as unbiased benchmarks for model evaluation.

Additionally, the Supporter Module enhances the model’s ability to capture and utilize contextual information, further boosting its performance in complex medical imaging tasks. The results demonstrate that RL-CancerNet outperforms existing SOTA models, achieving a remarkable 99.7% accuracy on benchmark datasets. This suggests that the proposed method not only advances automated cervical cancer screening but also holds potential for broader applications in other medical imaging domains. The findings of this study underscore the importance of incorporating advanced machine learning techniques, such as reinforcement learning and hybrid architectures, into models to improve the accuracy and reliability of medical diagnostics. Future work will focus on evaluating the model’s performance on more diverse and challenging datasets, as well as exploring its applicability to other areas of medical imaging, thus ensuring its robustness and generalizability across various clinical scenarios.

## Figures and Tables

**Figure 1 cancers-16-03782-f001:**
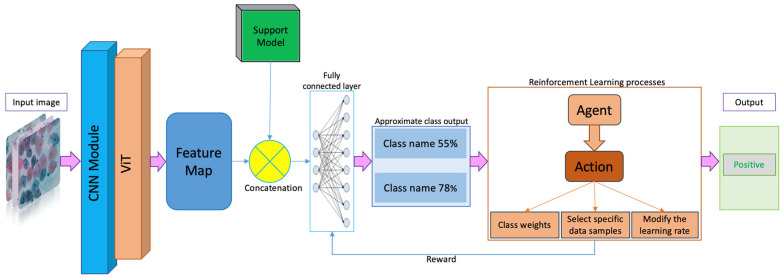
This image illustrates the workflow of the proposed reinforcement learning framework when integrated with a CNN–transformer model. The process includes feature extraction, patch processing, and dynamic adjustment by an RL agent to improve minority class classification accuracy.

**Figure 2 cancers-16-03782-f002:**
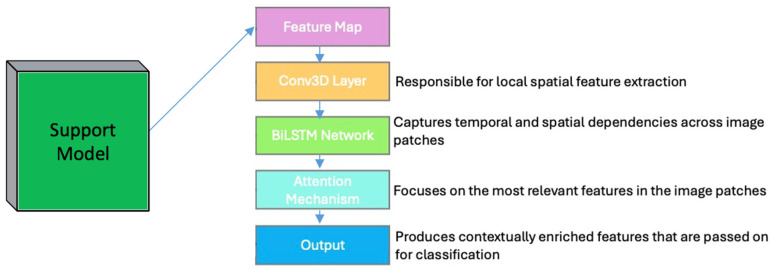
Depicts the architecture of the Supporter Module, which enhances contextual learning by capturing spatial and temporal dependencies in cervical cancer images using Conv3D, BiLSTM, and an attention mechanism.

**Figure 3 cancers-16-03782-f003:**
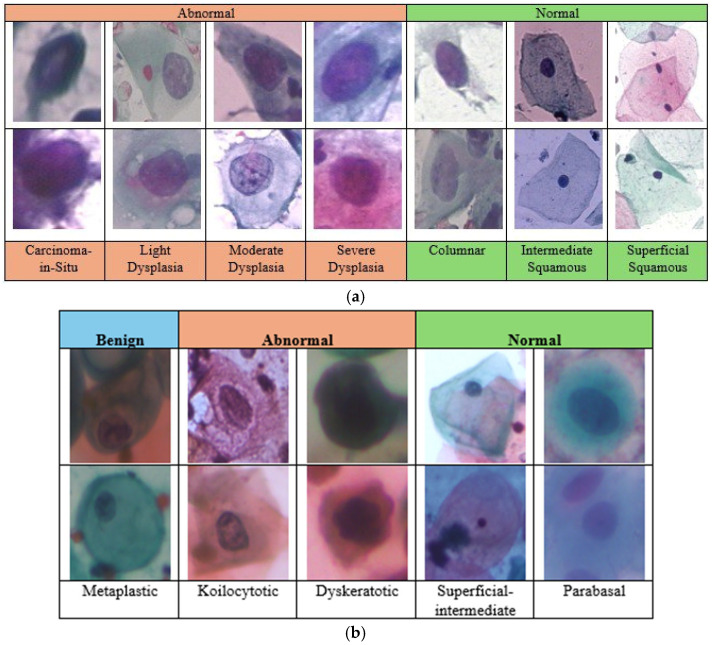
Description of cervical cancer datasets: (**a**) Herlev and (**b**) SipaKMed.

**Table 1 cancers-16-03782-t001:** Overview of data distribution in the SipaKMed dataset.

Category Name	Quantity	Property	Training Set	Validation Set	Test Set
Superficial–intermediate	813	Normal	569	122	122
Parabasal	787	Normal	551	118	118
Koilocytotic	825	Abnormal	578	124	123
Dyskeratotic	813	Abnormal	569	122	122
Metaplastic	793	Benign	555	119	119
Total	4049		2822	605	604

**Table 2 cancers-16-03782-t002:** Overview of data distribution in the Herlev dataset.

Category Name	Quantity	Property	Training Set	Validation Set	Test Set
Moderate squamous non-keratinizing dysplasia	146	Abnormal	102	22	22
Squamous cell carcinoma in situ intermediate	150	Abnormal	105	23	22
Severe squamous non-keratinizing dysplasia	197	Abnormal	138	29	30
Mild squamous non-keratinizing dysplasia	182	Abnormal	127	28	27
Columnar epithelial	98	Normal	69	15	14
Superficial squamous epithelial	74	Normal	52	11	11
Intermediate squamous epithelial	70	Normal	49	11	10
Total	917		642	139	136

**Table 3 cancers-16-03782-t003:** Comparison of classification performance using different models.

Metric	ResNet50 [24]	DeiT [25]	ViTs [10,26]	EfficientNetV1 [27]	ConvNext [28]	MobileNetV2 [29]	Inception V2 [30]	Proposed Method
Accuracy	96.24	97.00	94.44	95.21	96.54	95.21	96.21	98.89
Precision	95.25	96.10	96.87	94.11	96.21	94.00	96.00	98.99
Recall	97.12	96.02	97.00	93.27	97.00	95.64	96.05	98.32
F1	95.22	96.32	96.87	93.87	97.02	95.18	96.84	98.99

**Table 4 cancers-16-03782-t004:** Summarizing the computational efficiency and scalability of RL-CancerNet compared with simpler CNN-based models.

Model Type	Components	Memory Footprint	Computational Demand	Scalability in Clinical Settings
RL-CancerNet (Hybrid)	EfficientNetV2 + ViTs + RL	High (due to ViT layers)	High (quadratic scaling in ViTs)	Suitable for advanced hardware; limited real-time scalability without optimizations.
Pure CNN-based models	ResNet, VGG	Moderate to low	Moderate	High scalability; feasible for real-time applications on standard clinical hardware.
Optimized hybrid models	Swin transformer + CNN	Moderate	Moderate	Better suited for large datasets; can balance performance and scalability with optimizations.

**Table 5 cancers-16-03782-t005:** Comparison with DL methods.

Models	Accuracy	Precision	Recall	F1	Sens	Spec
CerviFormer [31]	97.03	96.23	97.62	94.25	97.5	67.8
CNN-based [32]	96.90	95.92	96.28	96.35	99.23	67.9
MaxViT ConvNeXt Based [33]	99.04	99.25	99.49	99.14	98.99	67.52
CNN + ViT [34]	98.25	97.85	98.07	97.01	98.22	67.18
Hybrid CNN [35]	99.56	99.24	98.99	99.06	98.56	67.25
Proposed	99.82	99.79	99.90	99.97	99.85	72.58

**Table 6 cancers-16-03782-t006:** Statistical comparison of performance metrics for RL-CancerNet and SOTA models.

Model	Accuracy (%)	Precision (%)	Recall (%)	F1 Score (%)	*p*-Value	Confidence Interval (95%)
ResNet50	98.52	97.31	98.38	97.7	<0.05	[0.85, 1.2]
Xception	96.16	98.15	99.21	98.36	<0.05	[1.1, 1.5]
EfficientNetV1	97.41	96.16	95.63	95.47	<0.05	[1.0, 1.4]
VGG16	96.23	95.12	95.6	95.28	<0.05	[0.9, 1.3]
MobileNetV2	97.55	96.99	97.82	97.25	<0.05	[0.8, 1.2]
InceptionV2	98.28	98.14	98.9	98.7	<0.05	[1.2, 1.6]
RL-CancerNet	99.7	99.36	99.9	99.72	N/A	N/A

## Data Availability

All datasets used are available online in an open access format.

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
