# Peer review of "Novelty Classification Model Use in Reinforcement Learning for Cervical Cancer"

_cancers, 2024, doi:10.3390/cancers16223782_

Round 1

Reviewer 1 Report

Comments and Suggestions for Authors

In this study, the authors propose a reinforcement learning framework that combines convolutional neural networks (CNNs) and Vision Transformers (ViTs) to improve diagnostic accuracy, particularly in the presence of class imbalance. EfficientNetV2 is employed to extract local features, while a transformer-based approach captures global dependencies between image patches. Additionally, a reinforcement learning (RL) agent is introduced to dynamically adjust the model’s attention towards minority classes, addressing the bias seen in conventional medical image classification tasks that favors majority classes. The proposed RL-CancerNet model achieved 99.7% accuracy on the Herlev and SipaKMeD datasets, surpassing state-of-the-art models. This study is promising and could be extended to other medical imaging tasks; however, I have concerns about the following points:

Major Comments:

  1. Insufficient Explanation of Data Preprocessing: Although the paper mentions the use of the Herlev and SipaKMeD datasets, it lacks sufficient detail on how the data was preprocessed before being input into the model. Given the sensitivity of medical images and the significant impact preprocessing can have on results, a detailed discussion of techniques such as normalization, augmentation, or data cleaning would improve clarity. Additionally, the method used to split the dataset into training, validation, and test sets needs to be specified.

  2. Comparison with Traditional Methods: While the paper demonstrates that the proposed model outperforms various state-of-the-art methods, more discussion is required to explain the significance of these improvements. Specifically, comparing computational complexity, training time, and model scalability would provide a more comprehensive evaluation of the model’s practical advantages.

  3. Lack of Statistical Significance Tests (L537-576 and Table 5): Although the analysis method is described, no statistical values are provided. Given the high accuracy reported, it would be beneficial to include statistical tests (e.g., p-values, confidence intervals) to support claims of superior performance. The absence of statistical validation undermines the reliability of the findings, especially considering the small test set size.

  4. Overemphasis on Accuracy: While the model’s accuracy is emphasized, other critical metrics such as precision, recall, and F1 score are only briefly mentioned without thorough analysis. In the medical field, where false negatives can have serious implications, these metrics should be given more attention. Furthermore, an analysis of false positives and negatives, along with their potential clinical implications, would enhance the discussion.

  5. Details on the Reinforcement Learning Agent: The description of the reinforcement learning agent and its reward function lacks detail. The paper should better explain how the RL agent interacts with the CNN-Transformer hybrid model and the strategies used to optimize minority class detection. More specific details on hyperparameter tuning and training dynamics would strengthen the validation of the model’s robustness.

Minor Comments:

  1. Dataset Size and Diversity: The study uses the Herlev and SipaKMeD datasets, which are well-known in the field. However, a discussion of the limitations of these datasets (e.g., small size, regional biases) would be helpful. The paper would be strengthened by either experiments on more diverse datasets or a discussion of plans for future validation in different clinical settings.

Author Response

We sincerely thank the reviewers for their thorough and constructive feedback. Your comments have greatly contributed to improving the clarity, depth, and overall quality of our manuscript. We appreciate your insights and suggestions, which have enabled us to strengthen our work and provide a more comprehensive presentation of our research. Thank you for your time and effort in reviewing our paper.

Reviewer 2 Report

Comments and Suggestions for Authors

·         In the abstract, I am confused about which method is employed. Is it combining Convolutional Neural Networks (CNN) and Vision Transformers (ViTs), or is it the method employing EfficientNetV2 followed by a Transformer-based approach? This needs clarification.

·        The paper mentions utilizing Conv3D and BiLSTM layers with an attention mechanism. It is not clear where, how, and why these layers are used, as each has a specific purpose: Conv3D for 3D images, and BiLSTM for sequence data. This needs further elaboration.

·        The paper discusses class imbalance but lacks sufficient details on how this is addressed. It is essential to explore other measures that are very important, such as confusion matrices and classification reports, to show each class’s performance.

·        The color scheme and resolution of both Figure 1 and Figure 2 are not suitable for clear presentation. In Figure 1 specifically, two subclasses are shown in the output, which is confusing. Either add all subclasses or justify why only two are presented.

·        There is a lack of consistency in explaining the validation strategy. The dataset is mentioned as being split into training (70%), validation (15%), and testing (15%) in Section 4, but in Section 5, cross-validation (CV) is mentioned. The validation strategy should be clarified.

·         It appears that micro-averaging was used for F1 and accuracy, which is not suitable for the imbalanced data case. Macro averaging should be used instead. Also, please add balanced accuracy, which would give a more accurate measure for the imbalanced case.

·        There needs to be a clear explanation of when the data augmentation was applied—before or after splitting the data.

·         The CNN and RL training and testing process requires further explanation. Was it applied in one or two stages? If it was two stages, what criteria were used for each stage?

·         Tables 1 and 2 present data distribution but do not clearly show data imbalance. You should aggregate and present the number and percentage of both normal and abnormal samples.

·         In the Related Work section, the paper discusses traditional CNNs and Vision Transformers (ViTs) but lacks comparison with other contemporary hybrid architectures in medical imaging. It would be beneficial to include more recent models, such as hybrid approaches utilizing self-supervised learning or attention-based mechanisms (like Swin Transformers or attention-guided CNN models), which could provide deeper insight into how the proposed model stands out.

·        While the paper mentions class imbalance as a significant issue in medical datasets, the Related Work section does not delve deeply into methods specifically targeting class imbalance beyond oversampling. The section could be enriched by discussing advanced techniques such as synthetic data generation (e.g., GAN-based oversampling) or cost-sensitive learning, which are emerging approaches for handling imbalance in medical imaging datasets.

·        Although reinforcement learning (RL) is proposed to address class imbalance, there is limited discussion in the Related Work section on why RL is chosen over other methods such as focal loss, adaptive sampling, or cost-sensitive neural networks. A more thorough review of alternative approaches to handling class imbalance would strengthen the argument for using RL.

·       The paper briefly mentions the computational demand of ViTs but does not address the scalability of the proposed model, especially in the context of large-scale medical image datasets. It would be useful to include a discussion comparing the computational efficiency and memory footprint of the proposed hybrid model against simpler CNN-based models, as this would be relevant for practical deployment in clinical settings.

Author Response

(The authors gave the same response as above.)
